# Body odor disgust sensitivity (BODS) is related to extreme odor valence perception

**Marta Zakrzewska**[1]*, **Marco Tullio Liuzza**[2], **Jonas K. Olofsson**[3]

**1** Department of Clinical Neuroscience, Karolinska Institutet, Solna, Sweden, **2** Department of Medical and Surgical Sciences, "Magna Graecia" University of Catanzaro, Catanzaro, Italy, **3** Department of Psychology, Stockholm University, Stockholm, Sweden

* marta.zakrzewska@ki.se

## Abstract

Odors are important disease cues, and disgust sensitivity to body odors reflects individual differences in disease avoidance. The body odor disgust sensitivity (BODS) scale provides a rapid and valid assessment of individual differences. Nevertheless, little is known about how individual differences in BODS might correlate with overall odor perception or how it is related to other differences in emotional reactivity (e.g., affect intensity). We investigated how BODS relates to perceptual ratings of pleasant and unpleasant odors. We aggregated data from 4 experiments (total N = 190) that were conducted in our laboratory, and where valence and intensity ratings were collected. Unpleasant odors were body-like (e.g., sweat-like valeric acid), which may provide disease cues. The pleasant odors were, in contrast, often found in soap and cleaning products (e.g., lilac, lemon). Across experiments, we show that individuals with higher BODS levels perceived smells as more highly valenced overall: unpleasant smells were rated as more unpleasant, and pleasant smells were rated as more pleasant. These results suggest that body odor disgust sensitivity is associated with a broader pattern of affect intensity which causes stronger emotional responses to both negative and positive odors. In contrast, BODS levels were not associated with odor intensity perception. Furthermore, disgust sensitivity to odors coming from external sources (e.g., someone else's sweat) was the best predictor of odor valence ratings. The effects were modest in size. The results validate the BODS scale as it is explicitly associated with experimental ratings of odor valence.

## 1. Introduction

Disgust is important for behavioral disease avoidance [1]. Disgust can arise in response to pathogen threats or pathogen-resembling cues [2] and it leads to avoidant reactions. Due to its association with behavioral avoidance, disgust sensitivity may be involved in shaping social attitudes [3]. Odors, and body odors in particular, convey important pathogen and health-related information [4–6], and olfactory cues are potent triggers of disgust and avoidance responses [7, 8]. Disgust response to olfactory stimuli is less prone to habituation than response to similar visual, auditory, or tactile stimuli, and evokes a different autonomic reaction [9], highlighting how relevant the sense of smell is for disgust. Olsson et al. [4] showed

**Data Availability Statement:** All data and scripts are available at OSF (https://osf.io/c7w84/?view_only=edd70587dad84109a1f3cb950aefbd08).

**Funding:** This work has been supported by the research grants from the Swedish Research

Council (2016-02018) to M.T.L and Knut and Alice
Wallenberg Foundation (2016:0229) to J.K.O. The
funders had no role in study design, data collection
and analysis, decision to publish, or preparation of
the manuscript.

**Competing interests:** The authors have declared
that no competing interests exist.

that sweat samples from sick individuals were rated differently in terms of perceptual ratings:
'sick' sweat was more intense and less pleasant than healthy sweat. Previous research shows
that disgust sensitivity to body odors reflects individual differences in disease avoidance and
may be an aspect underlying some social attitudes [10]. The body odor disgust sensitivity
(BODS [11]) scale provides a rapid and valid assessment of individual differences.

The relationship between individual differences in disgust sensitivity and the perception of
health/disease-relevant cues has not been thoroughly studied. In the visual domain, higher dis-
gust sensitivity was positively related to disgust ratings of disgusting images [12, 13]. As for
olfactory stimuli, one study found that higher general disgust sensitivity was related to lower
pleasantness ratings for an unpleasant, feces-like smell, but not to the perception of the pleas-
antness of several other odors; nor was it related to the perception of the intensity of any of the
smells [14]. Body odor disgust, as measured by the BODS scale, is positively related to ratings
of disgust, but not intensity, of armpit sweat biosamples [15]. Croy et al. [16] found a positive
relationship between olfactory sensitivity (e.g., performance on olfactory threshold task) and
overall disgust sensitivity in men, while other groups found no relationship between the two
([17–19], although [19] had only female participants). Unfortunately, neither study looked at
the relationship between trait disgust sensitivity for odors and the perception of the study
odors. It is thus not yet clear how trait disgust sensitivity may be reflected in odor perception.
The present study aims to fill this gap.

In the current study, we investigated if disgust sensitivity to body odors corresponds to per-
ceptual ratings of both pleasant (e.g., safety, cleanness, or health-related) and unpleasant (e.g.,
potentially threatening, disease-relevant) odors. We included odors that might resemble or be
associated with 'real' body odors, or fragrant products used on the body. As disease and health
cues in odors such as human sweat seem to have an effect on the perception of intensity and
pleasantness of the odor [4], we addressed the question of whether levels of odor disgust sensi-
tivity would be related to perceptual odor ratings. We approached the topic by combining data
from four experiments conducted in our lab, in which perceptual odor ratings were collected.
This allowed us to have a larger and more varied data set and a potentially better estimation of
effects in the data, in the sense that they were less dependent on a particular stimulus or subset
of individuals.

The BODS scale has two subscales: the internal (odor source is one's own body) and exter-
nal (odor source is a stranger's body). The two subscales are highly correlated ($r = 0.67$) and
both are associated with disease avoidance traits [11]. Nevertheless, they might be related to
different aspects of disease avoidance: detecting disease cues from odors emitted by other peo-
ple, and avoiding these people, can help prevent catching a disease, while monitoring one's
own body status via odors, can help trigger relevant disease-coping behaviors (sickness behav-
iors) that channel the energy appropriately and maximize the efficiency of the immune system
[20, 21]. Interestingly, a recent study showed that over 90% of participants reported smelling
themselves, suggesting that our own body odor is an important and frequently accessed signal
[22]. If people smell themselves often, then maybe reactions to internal sources of odors
deserve more attention in disease avoidance and other contexts. Additionally, reactions to
one's own smell are potentially less normative and culture-dependent as compared to
responses to external sources of body odors. To sum up, the internal and external subscales of
BODS might provide complementary information regarding disease avoidance.

We hypothesized that there would be a relationship between BODS levels and perception of
odors. This relationship could take one of several forms, each being informative about the
validity of the BODS scale and the nature of individual differences in disgust sensitivity. First,
high BODS levels might be related to perceiving odors as generally more intense (Scenario 1),
similar to what we see in people with a high environmental chemosensory responsivity, who

are disturbed by perfume, cigarette smoke, etc. [23]. This alternative seems rather unlikely, as a previous study showed that BODS is related only to valence ratings, but not intensity ratings, of human armpit sweat [15]. Another hypothesis (Scenario 2) is that people high in BODS will perceive all smells as more unpleasant: the disgust sensitivity would thus be driven by the negative experience of smelling odors in general, which would point to a heightened sensitivity to negative aspects of the olfactory environment. We note that this tendency would not be highly adaptive, as reacting negatively to pleasant or health-related odors would be impractical in avoiding diseases. This leads us to two other potential outcomes, both of which seem more likely: BODS may be related to a more negative perception specifically for potentially hazardous (i.e., unpleasant) odors (Scenario 3). A negative relationship with valence ratings for unpleasant odors, but not for pleasant ones, would suggest that disgust sensitivity is uniquely related to the perception of potentially pathogen-related cues, but not neutral or safety cues. This scenario would be in line with results from previous research [14]. Lastly, the relationship between BODS and odor valence may be different for pleasant and unpleasant odors: disease-related, unpleasant odors would be perceived as more negative, stimulating avoidance behaviors, while health or pleasant cues would be perceived as more positive since they provide an olfactory "safety cue" (Scenario 4). If any of the four possible scenarios mentioned above were confirmed, it would provide valuable information about the role of individual differences in body odor disgust sensitivity, assisting theoretical development. Additionally, by looking at the emerging scenario in the context of the two BODS subscales (body odors from self vs. others), we will be able to draw preliminary conclusions in terms of whether self-monitoring, reactivity to external signals, or a combination of both, is most relevant for odor perception.

## 2. Methods

We aggregated data from experiments conducted in our laboratory and fulfilled our inclusion criteria.

### 2.1. Experiment inclusion criteria

We included unpublished data from four experiments performed in our lab. The inclusion criteria were the following a) the studies had to have perceptual ratings of odors in at least two dimensions: intensity and pleasantness, b) the odor stimuli had to include both commonly pleasant and commonly unpleasant smells; and c) studies used the BODS questionnaire [11]. Although two of the four experiments were part of studies that were previously published [24, 25], the current ratings and/or survey data analyses we focused on here were not included in these prior publications. Table 1 contains information about the experiments included in this paper.

### 2.2. Participants

The final sample consisted of 190 participants across four experiments (see Table 1 and *2x.6 Data exclusion* for more information about the sample). All studies complied with the Declaration of Helsinki and the research was approved by the regional ethics board (2017/2277-32). Participants provided informed written consent.

### 2.3. Odor stimuli and perceptual ratings

All experiments included clearly valenced odors. Unpleasant odors were selected in order to provide disease cues: sweat-like (valeric acid), vomit-like (butyric acid), fecal-like (skatole) smells, and a compound found in breath (2-picoline). Pleasant stimuli included odors

**Table 1. Descriptive statistics and mean scores on the body odor disgust sensitivity (BODS) scale in each study.**

|  | N | Age |  | BODS | |
|---|---|---|---|---|---|
|  | (F) |  | Overall | Internal | External |
| EXP 1 | 53 | 25.8 | 3.0 | 2.4 | 3.6 |
|  | (29) | (5.8) | (0.6) | (0.7) | (0.6) |
| EXP 2 | 35 | 28.5 | 3.2 | 2.8 | 3.6 |
|  | (25) | (6.3) | (0.8) | (0.9) | (0.9) |
| EXP 3 | 22 | 28.0 | 3.1 | 2.5 | 3.7 |
|  | (15) | (6.2) | (0.6) | (0.7) | (0.8) |
| EXP 4 | 80 | 29.2 | 3.1 | 2.6 | 3.6 |
|  | (46) | (8.3) | (0.7) | (0.8) | (0.7) |

EXP–experiment; N–number of participants; F–number of female participants; Age–mean age (standard deviation). Overall–overall score on the BODS scale; Internal–score on the internal subscale of the BODS scale; External–score on the external subscale of the BODS scale.

associated with food, flowers, and cleaning products, and thus assumed to have positive health connotations: lilac, lemon, peppermint, and lily of the valley essences. In our previous work, lilac was found to be strongly associated with cleanness [26]. All odorant solutions were diluted in an odorless mineral oil solution (Propylene glycol, 1.2-propanediol 99%, Sigma-Aldrich). In experiments 1 [24] and 3 (unpublished), the odors were presented in a brown glass jar, several times in randomized order. After each presentation, participants were asked to rate the stimulus intensity and valence. In experiment 2, a drop of odor solution was placed on a medical cotton pad, which was then placed in tube-shaped cotton bands, and tied under the participant's nose. Participants were thus exposed to odors continuously over a period of several minutes while performing a task [25]. Perceptual ratings were done before and after the task. In experiment 4 (unpublished), smells were delivered using an olfactometer [27] and ratings were done immediately after odor exposure. The perceptual rating procedure was repeated twice, once before, and once after participants completed a priming odor paradigm using the same odor stimuli. Thus, the present study included odor-rating data collected in varying circumstances.

As the measures and scales used were not identical across all studies, we used only ratings common to all four experiments (pleasantness and intensity). We standardized the rating data within each experiment to deal with differences in scale ranges used. Table 2 provides a summary of methods, stimuli, and ratings used in each experiment.

## 2.4. Body odor disgust sensitivity scale

In all experiments, participants filled in the BODS [11] questionnaire using the online Qualtrics survey platform (Qualtrics, Provo, UT; experiments 2, 3, and 4) or on paper (experiment 1). BODS is a 12-item scale that measures disgust sensitivity to body odors. Items refer to six types of body odors (feces, upper body sweat, feet, urine, gas, and breath) appearing both in an internal (e.g., "You are alone at home and notice that your feet smell strongly") and external (e.g., "You are sitting next to a stranger and notice that their feet smell strongly") contexts. Participants rated the extent to which each scenario elicits disgust on a Likert-type scale ranging from 1 (*not disgusting at all*) to 5 (*extremely disgusting*). For each participant, we calculated the overall BODS mean score, and mean scores on the internal and external subscales (Table 1).

## 2.5. Data analysis

We aggregated data from all four experiments and performed Bayesian parameter estimation and modeling using multilevel models, to account for individual differences in the base level of

**Table 2. Information about smell and ratings used in each study.**

|  | Odors | Odor characteristics | Smell delivery | Ratings collected | Trials per odor | Rating scale (s) | Relevant publication |
|---|---|---|---|---|---|---|---|
| EXP 1 | **Lilac** *Valeric acid* | soap sweat | cotton pad with a drop of odor solution placed under the nose | **Pleasantness Intensity** | 3 | -10 to 10 1 to 120* | Syrjänen et al. (2018)** |
| EXP 2 | **Lilac** *Valeric acid* | soap sweat | cotton pad with a drop of odor solution placed under the nose | **Pleasantness Intensity** | 2 | 1 to 7 | Zakrzewska et al. (2020) |
| EXP 3 | **Lilac** Lily of the valley Peppermint *Valeric acid* *Scatole* *Butyric Acid* | soap soap, perfume fresh breath sweat feces vomit | a jar containing odor solution | **Pleasantness Intensity** Familiarity | 3 | 0 to 100 |  |
| EXP 4 | **Lilac** Lemon *Valeric acid* *2-picoline* | soap fresh sweat breath compound | olfactometer | **Pleasantness Intensity** Disgust | 2 | -100 to 100 1 to 120* 1 to 120* |  |

EXP–experiment; N–number of participants; F–number of female participants; Age–mean age (standard deviation). Italics indicate unpleasant smells; bold indicates smells and ratings present in all four experiments

* Borg CR-100 scale (Borg & Borg, 2002)

** data used here comes from the initial rating task, not ratings within the main task. See [24] for more details.

odor rating. We used R [28] via RStudio [29], and the *rethinking* package [30]. We modeled the ratings using regularizing priors and normal distribution (M = 0, SD = 0.5) for the intercept and beta coefficients, and an exponential distribution for the sigma parameter.

**2.5.1. Relationship between BODS and perceptual odor ratings.** In the first step of the analysis, we asked how overall BODS levels are related to the perception of odor valence and intensity. We created a set of models with relevant predictors to perform a model comparison, separately for the two outcomes: intensity and valence. As each individual in the dataset gave multiple ratings of each odor, all models allowed individual intercepts for overall ratings for each participant (random effect of participants' ID). As gender is known to affect both olfactory functions and disgust sensitivity, all models included the effect of gender, even though we were not specifically interested in gender differences. Our null model included only the effect of gender and the (random) effect of participant ID. The two simplest models also included either the effect of odor category or the effect of BODS. Next, we constructed a model with the additive effect of odor and BODS. Lastly, the most complex model included the additive effects of odor category and BODS, and the interaction between the two.

The best model was chosen based on the difference in information criteria (widely applicable information criterion (also known as the Watanabe–Akaike information criterion), WAIC; difference denoted as ΔWAIC), and the standard error of the difference (ΔSE). For a model to be considered better, it had to have a lower WAIC value than the alternative model(s), and this difference had to be at least twice as big as the ΔSE. The best model was then used to estimate the relevant effects. We provide coefficient estimates with 94% (see [31] p. 56 for recommendations against using 95%) highest posterior density interval (HPDI). The model comparison and the estimates from the best model allowed us to decide whether BODS is related to perceptual ratings, and in what way.

These models allowed us to test scenarios described in the introduction. For example, if the scenario outlined as Scenario 1 was true, an additive model including BODS and odor

category, or a simple model with BODS effect only, should have the best fit to the intensity data. There should be a general (positive) relationship between intensity and BODS (regardless of whether or not the two odor categories are rated as having a similar intensity). Alternatively, Scenario 2 would be indicated if an additive model of BODS and odor category provided the best fit to the valence data (odors should be rated differently based on odor category, but the effect of BODS should be similar for both categories). Scenarios 3 and 4 would be indicated if an interaction model came out best for valence ratings, with either a (negative) BODS effect visible only for unpleasant odors (Scenario 3) or for both odor categories, in the opposite direction (negative for unpleasant, positive for pleasant; Scenario 4). Lastly, suppose our hypothesis that there is a relationship between odor ratings and BODS was false. In that case, a simple model that included only the odor category should provide the best model for both intensity and valence ratings. This list does not exhaust all possible outcomes of the model comparison, yet they correspond to the theoretically plausible options we had hypothesized.

**2.5.2. Different aspects of BODS: the internal and external subscales.** In the second step, we asked which aspect of BODS is the best predictor of odor ratings: a) overall BODS score (used in step 1), b) internal BODS subscale score, c) external BODS subscale score, or d) both internal and external subscale scores (included separately, not as an overall score). For this step, we re-estimated the best model from step one using the other BODS indices (b–d) instead of overall BODS. Additionally, we had two 'intermediate' models to allow for a clear comparison between models, including two subscales and models with only one subscale (or the overall score). These two intermediate models included the main effects of the two subscales and the interaction between the odor category and either the internal (intermediate model 1) or external (intermediate model 2) subscales. These two models were necessary to ensure there is only one parameter difference between subsequent models (e.g., between internal subscale interaction model vs intermediate model 1-, or between intermediate model 1 vs. model with two interactions). We applied the same model comparison strategy as in step one. This allowed us to assess which aspect of BODS has the best predictive value for perceptual odor ratings.

**2.5.3. Exploratory analysis of gender differences.** As an additional step, investigated possible gender differences in the effects that of odor category and BODS have on pleasantness and intensity ratings. These extra analyses are described in detail in S1 Appendix.

## 2.6. Data exclusion and missing data

The initial sample size from all four experiments consisted of 196 participants. Three participants were excluded from experiment 4 as they quit the main experiment task due to the odors being too overwhelming. Additionally, we excluded 3 (one from experiment 3 and two from experiment 4) raters that seemingly misused the scale: these participants rated unpleasant smells as very pleasant and vice versa. We assumed these participants confused (flipped) the scale ends when providing the ratings and decided these should not be used for making inferences. For an individual to be removed, two requirements needed to be met: participants had to rate the pleasant smells below 30% of maximum pleasantness and unpleasant smells above 70% of maximal pleasantness. After removing participants due to incorrect use of the rating scales, the final sample size included 190 individuals. Demographics in Table 1 are given for this final sample size, after applying all exclusion criteria. Additionally, we would like to note that 5 participants from the original sample in experiment 1 were not included in the initial sample size due to missing BODS scale data, which explains the slight difference in N reported here and in [24].

**Table 3. Mean (standard deviation) odor ratings of pleasantness and intensity in each experiment (EXP), separately for two odor categories (pleasant and unpleasant).** Text in italics indicates the range of the scales used. To ease the interpretation of the ratings done on different scales, we transformed the raw scores into proportions of the maximal value on each scale (prop. max). * Borg CR-100 scale [32]. In this scale, the maximal value is 100, but participants can go beyond the maximal value (up to 120).

| | Odor category | Pleasantness | | (prop. max) | Intensity | | (prop. max) |
|---|---|---|---|---|---|---|---|
| EXP1 | Pleasant | *-10 to 10* | 4.8 (2.6) | 0.74 (0.13) | *1 to 120** | 34.3 (17.2) | 0.34 (0.17) |
| | Unpleasant | | -3.3 (3.1) | 0.33 (0.16) | | 32.8 (20.3) | 0.33 (0.20) |
| EXP2 | Pleasant | *1 to 7* | 5.8 (0.9) | 0.83 (0.13) | *1 to 7* | 5.6 (0.9) | 0.8 (0.13) |
| | Unpleasant | | 2.8 (1.2) | 0.4 (0.17) | | 5.8 (1.0) | 0.83 (0.14) |
| EXP3 | Pleasant | *0 to 100* | 73.6 (20.7) | 0.74 (0.21) | *0 to 100* | 60.5 (21.6) | 0.61 (0.22) |
| | Unpleasant | | 18.9 (19.0) | 0.19 (0.19) | | 66.6 (24.0) | 0.67 (0.24) |
| EXP4 | Pleasant | *-100 to 100* | 35.3 (40.0) | 0.68 (0.20) | *1 to 120** | 37.7 (20.2) | 0.38 (0.20) |
| | Unpleasant | | -43.3 (42.3) | 0.28 (0.21) | | 53.8 (26.9) | 0.53 (0.26) |

## 2.7. Data transformation

As ratings were not done on the same scales in all experiments, we standardized the data within each experiment.

## 3. Results

### 3.1. Valence ratings

Mean pleasantness ratings (raw) from each study are shown in Table 3. To ease the comparison of ratings done on different scales, we also translated the raw ratings into a proportion of maximal value on each scale.

**3.1.1. Individuals with higher BODS levels perceive pleasant smells as more pleasant, and unpleasant smells as more unpleasant.** Model comparison indicated an interaction between the odor category and BODS: the interaction model was better than the additive model and better than models including only the odor category or only BODS (Table 4). In the best (interaction) model, unpleasant smells were rated as less pleasant than pleasant smells (-0.7 [-0.94–0.47], Fig 3). Results supported one of our hypothesized outcomes, as individuals with higher BODS levels rated odors as more strongly valenced: unpleasant smells were rated as more unpleasant, but pleasant smells were rated as more pleasant (Fig 1). The smells were rated somewhat more pleasant by women than by men (0.16 [0.03 0.29]). Exploratory analysis suggests that this gender difference might apply mostly to pleasant odors (see S1 Appendix, *2.1. Gender differences, body odor disgust sensitivity and valence ratings*)

**Table 4. Model comparison for models predicting valence ratings.**

| | WAIC | ΔWAIC | Δ SE | ΔWAIC /Δ SE |
|---|---|---|---|---|
| **Odor * BODS overall** | **2675** | **0.0** | - | - |
| Odor + BODS overall | 2703 | 28.0 | 13.7 | 2.0 |
| Odor | 2704 | 29.6 | 13.8 | 2.1 |
| Null model | 4246 | 1571.7 | 69.7 | 22.5 |
| BODS | 4247 | 1572.6 | 69.7 | 22.6 |

Models are presented based on their WAIC value: lowest (i.e., better; top row) to highest (i.e., worse; bottom row), and the best model is marked in bold. WAIC—widely applicable information criterion; Δ WAIC–WAIC difference (vs. model in the top row); Δ SE—standard error of the WAIC difference. BODS overall–body odor disgust sensitivity overall score. The null model included gender and random intercepts for each participant
* Indicates an interaction effect and + an additive effect.

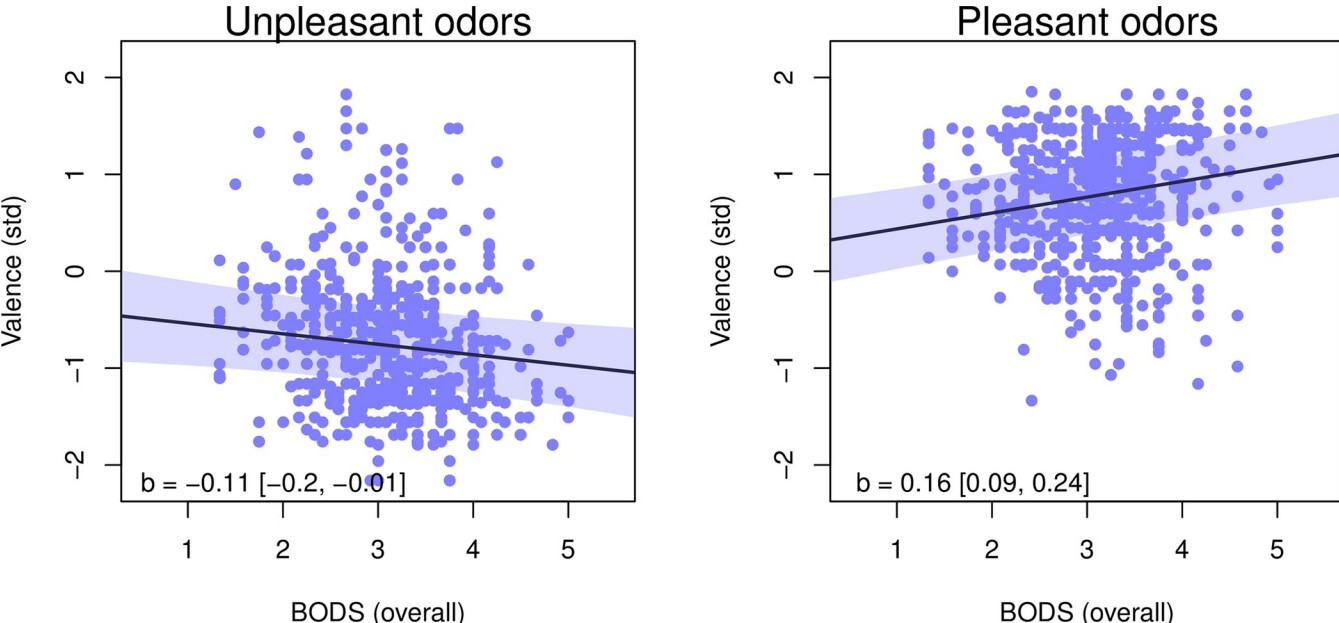

**Fig 1.** Relationship between overall BODS score and standardized valence ratings for unpleasant (left) and pleasant (right) odors. The shaded area and numbers in square brackets represent 94% HPDI for the slope.

**3.1.2. BODS external subscale predicts odor valence best.** Three models had similar WAIC values and were thus fairly indistinguishable ($\Delta$WAIC $< = 1$, $\Delta$SE $> = \Delta$WAIC). All three models included the interaction between the odor category and external BODS subscale, and the most parsimonious model should be preferred. This model included the interaction of the odor category and external BODS subscale. Thus, disgust sensitivity to odors coming from external sources (e.g., stranger's sweat) was considered the best predictor of odor valence among all the BODS indices. The model with interaction between odor category and external BODS score performed better than interaction models with internal or overall BODS (Table 5).

**Table 5. Model comparison for models predicting valence ratings.**

|  | WAIC | Δ WAIC | Δ SE | ΔWAIC /Δ SE |
|---|---|---|---|---|
| **Odor * BODS external** | **2656** | **0.0** | - | - |
| Odor * BODS external + BODS internal | 2656 | 0.6 | 0.6 | 1.0 |
| Odor * BODS external + Odor * BODS internal | 2657 | 0.9 | 3.6 | 0.2 |
| Odor * BODS overall | 2675 | 18.9 | 7.8 | 2.4 |
| Odor * BODS internal + BODS external | 2696 | 40.4 | 13.4 | 3.0 |
| Odor * BODS internal | 2699 | 43.3 | 13.3 | 3.3 |

Models are presented based on their WAIC value: lowest (i.e., better; top row) to highest (i.e., worse; bottom row), and the best model is marked in bold. WAIC–widely applicable information criterion; Δ WAIC–WAIC difference (vs. model in the row below); Δ SE–standard error of the WAIC difference. BODS–body odor disgust sensitivity computed as the overall score (BODS overall), the score on the internal subscale (BODS internal), and the score on the external subscale (BODS external). All models with an interaction term also included corresponding main effects of odor category and BODS. Two models were intermediate, including the corresponding BODS main effect, and an additional main effect of the other BODS subscale, as specified by the + term

* Indicates an interaction effect and + an additive effect.

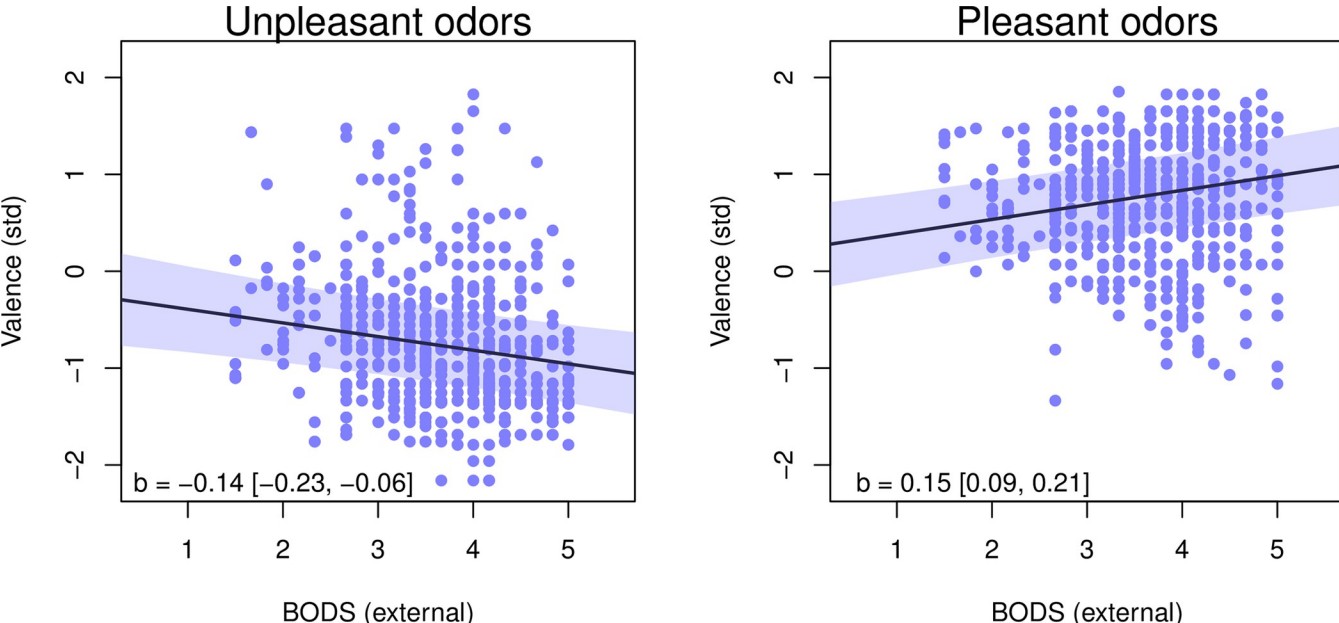

**Fig 2.** Relationship between external BODS and standardized valence ratings for unpleasant (left) and pleasant (right) odors. The shaded area and numbers in square brackets represent 94% HPDI for the slope.

The results from the external BODS model were similar to those when using overall BODS: again, unpleasant smells were rated as less pleasant than pleasant smells -0.49 [-0.73–0.24]) and individuals with higher BODS levels perceived pleasant smells as more pleasant overall: unpleasant smells were rated as more unpleasant, but pleasant smells were rated as more pleasant (Fig 2). Again, women rated the smells somewhat more pleasant than men (0.14 [0 0.27]).

## 3.2. Intensity ratings

Mean intensity ratings (raw) from each study are shown in Table 6. As in the case of pleasantness ratings, we additionally translated the raw ratings into a proportion of maximal value on each scale.

**3.2.1. BODS levels do not strongly affect the perception of odor intensity.** Although the model with BODS and odor category interaction had the lowest WAIC, all models

**Table 6. Model comparison for models predicting intensity ratings.**

|  | WAIC | ΔWAIC | Δ SE | ΔWAIC /Δ SE |
|---|---|---|---|---|
| Odor * BODS overall | 3706 | 0.0 | - | - |
| **Odor** | **3709** | **3.1** | **6.6** | **0.5** |
| Odor + BODS overall | 3709 | 3.3 | 6.7 | 0.5 |
| Null model | 3775 | 68.7 | 17.7 | 3.9 |
| BODS | 3776 | 69.7 | 17.7 | 3.9 |

Models are presented from lowest (top row) to highest WAIC value (bottom row) and the best model is marked in bold. WAIC—widely applicable information criterion; Δ WAIC–WAIC difference (vs. model in the top row); Δ SE—standard error of the WAIC difference. BODS overall–body odor disgust sensitivity overall score. The null model included gender and random intercepts for each participant

* Indicates an interaction effect and + an additive effect.

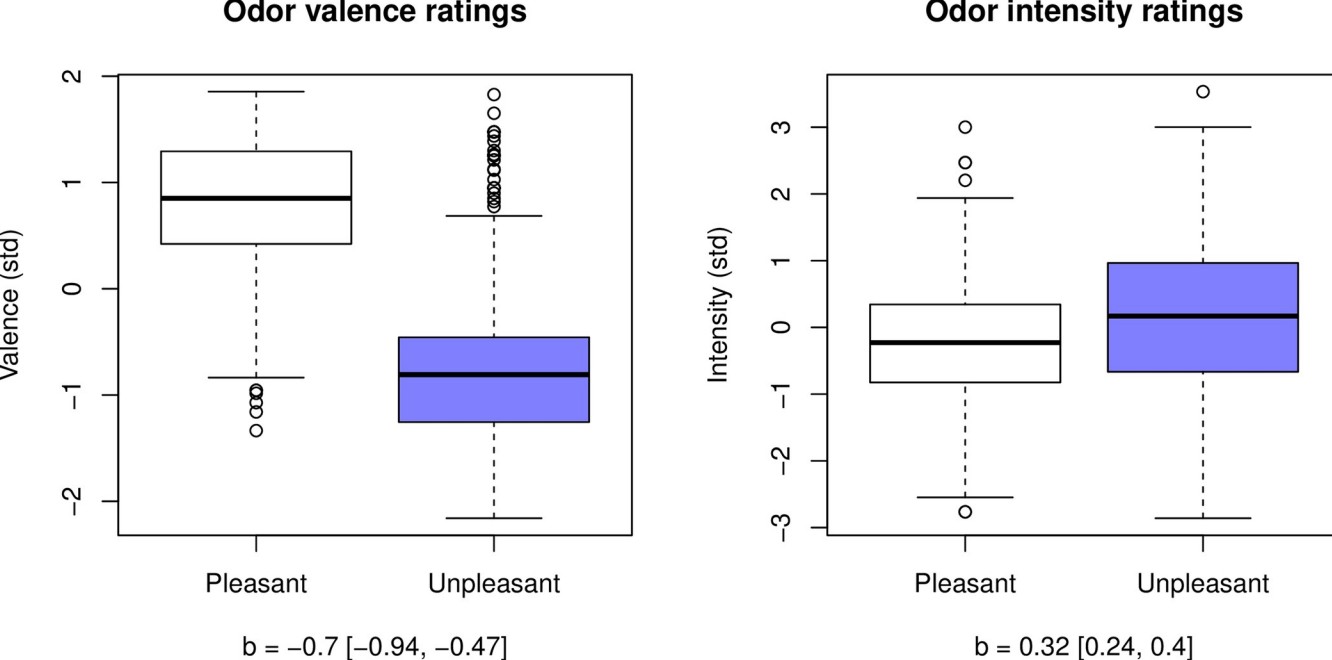

**Fig 3.** Standardized valence (left) and intensity (right) ratings for pleasant (white) and unpleasant (blue) odors. Black lines show medians, numbers in square brackets represent 94% HPDI for mean posterior estimate of the difference (b).

including the odor category performed similarly (all $\Delta$WAIC < $\Delta$SE, Table 6), and thus the most straightforward model (odor category effect only) should be favored because it is the most parsimonious. In this simple (odor-only) model, unpleasant smells were perceived as more intense by 1/3 of a standard deviation (0.32 [0.24 0.4], Fig 3). Smells were rated somewhat less intense by men than women (-0.09 [-0.14–0.03]). The exploratory analysis did not provide any more insight into gender differences (see S1 Appendix, *2.2. Gender differences in intensity ratings*)

As BODS was not clearly related to intensity ratings, we did not proceed to the next step of our analysis (looking at different aspects of BODS). We thus concluded the investigation of intensity ratings at this stage.

## 4. Discussion

Disgust sensitivity to body odors varies in the population and it might reveal a broader set of individual differences relating to emotions, perception, and even social attitudes and preferences for limited intergroup contact and social mobility [11, 25]. In this context, studying body odor disgust sensitivity can provide useful information about disease avoidance and the relevant behaviors. However, our insights into body odor disgust sensitivity (BODS) are mostly based on non-experimental survey data. Although previous research showed that BODS is associated with higher disgust ratings when smelling armpit sweat biosamples [15], there has been a gap in the literature regarding how BODS is associated with odor perception more generally. In fact, there is not much research regarding how disgust sensitivity translates to the perception of disgusting (and not disgusting) cues. The present results explain how individual differences in BODS are associated with the perception of odor valence and intensity in an experimental setting. Individuals who are easily disgusted by body odors perceived odor valences more strongly: pleasant odors were rated as more pleasant, and unpleasant odors

were rated as more unpleasant. The intensity of the odor, however, was perceived similarly by participants across all levels of BODS. This finding is in line with our previous results from sweat biosamples [15], and it effectively rules out the interpretation that general differences in using the rating scales would somehow explain our correlations. Interestingly, Olsson et al. [4] show that when rating sick vs healthy human sweat samples, disease-associated sweat is rated as both more unpleasant and more intense, even if the differences in unpleasantness were slightly more pronounced *(Cohens d* for intensity = 0.21 vs d = 0.26 for pleasantness). Our present results replicate and extend the results of Liuzza et al. [15], where participants with high levels of BODS rated human armpit sweat as especially disgusting, but not as especially intense. Taken together, these results suggest that disgust sensitivity to body odors is related more to the qualitative aspects of the smell signal (pleasantness, disgustingness) rather than the strength of the signal itself (intensity). We speculate that disease avoidance is not only related to avoidance behaviors (responding more negatively to potentially threatening stimuli) but also to the initiation of approach behaviors (by perceiving pleasant and potentially health-related stimuli more positively). Across our current experiments, individuals who are high in BODS demonstrate both of these traits.

Our results are in line with the affect intensity theory [33], which states that some individuals will show greater emotional experiences, both for positive and negative emotions. Here, individuals more sensitive to experiencing disgust showed both more negative reactions to negative stimuli as well as more positive reactions to positive stimuli. Thus, this study allows us to place BODS in a broader context, incorporating a key concept from the broader emotion literature where olfactory perception is rarely considered.

Our results also converge with a recent study by Tybur and colleagues [34] who found a relationship between disgust sensitivity (as measured by several disgust measures, including BODS) and perception of unpleasant, pathogen-relevant smells.

It may be useful to interpret our results from a signal detection theory (SDT) perspective whereby individuals differentiate environmental cues from noise by applying a response criterion. In the light of SDT, individual disgust sensitivity should depend on two parameters: sensitivity/easiness to detect a disgusting odor (i.e., the detection threshold, referred to as $d'$) and sensitivity/easiness to call an odor disgusting (i.e., the criterion). As levels of BODS affected valence ratings but not intensity ratings, we hypothesize that people with higher BODS levels are more likely to have a lower criterion: they more easily judge an unpleasant odor as unpleasant (or a pleasant odor as pleasant) rather than detect the odor itself more easily. Put differently, what differentiates individuals who are highly disgust-sensitive to body odors from those who are insensitive, are likely central processing factors related to emotion and cognition rather than a heightened olfactory sensitivity. Accordingly, previous research suggested no relationship between disgust sensitivity and olfactory thresholds [17, 19], although the results are somewhat inconsistent (see [16] for a divergent result). Our results and interpretations are in line with recent results from research on visual disgust. When studying how disgusting images can cause an attention blink effect, researchers showed that individual differences in disgust sensitivity were related to the valence ratings of the image causing the effect to appear, rather than the attention blink effect itself [35]. Similarly, [34] found qualitative (perception of pleasantness) but not quantitative (detection thresholds) effects of disgust sensitivity on odor perception. However, a limitation of our study is that we do not test the threshold of detecting an odor directly. Thus, our hypotheses concerning lower criterion vs. lower threshold are preliminary and should be tested directly in future studies.

Our study results further validate the BODS scale, a quick and easy tool to measure individual differences in disgust sensitivity to body odors. Body odors are difficult to sample and store, and the BODS scale can reliably approximate people's reactions to them. Our study adds

to previous research, showing that BODS is related to perceptual ratings of not only armpit sweat but also to various pleasant and unpleasant non-body odors. We used four different pleasant and four unpleasant smells in our four different studies. The results observed in the combined dataset were highly similar in each study (see S1 Fig in S1 Appendix). This allows us to say more about the generalizability of the BODS–valence rating relationship than if we had used data from one study only. Additionally, our study is among the few that have tried to relate self-reported disgust sensitivity to the perception of disgusting stimuli.

Our study shows that the external subscale of BODS is the best predictor of ratings of odor pleasantness. Again, this aligns with previous research using sweat biosamples [15]. From a disease avoidance perspective, the external subscale would be the most relevant for avoiding interaction with potentially contagious individuals. Perceiving unpleasant odors as even more unpleasant would surely boost one's will to avoid the source of the smell. Similarly, perceiving pleasant odors as more pleasant would be a stronger safety cue and could foster approach behaviors. Both relate to proactive behaviors that balance the cost-benefit trade-off of interacting with other people. Thus, it is unsurprising that the external scale is a good predictor of valence ratings.

The scope of this study was limited to intensity and pleasantness ratings only, and the stimuli were clearly coming from an external source. However, other perceptual aspects of odor perception might be more strongly related to the internal subscale of BODS. Being more or less sensitive to odor signals coming from one's body can have affect judging one's health and adjusting our behaviors accordingly.

## 5. Conclusions

We show that self-reported disgust sensitivity to body odors is related to the perception of odors, namely odor valence, suggesting that BODS is associated with both *negative* and *positive* reactions to odor cues. This work is a step further in understanding olfactory disgust's role in disease avoidance. More research is needed to make firm conclusions about the underlying mechanisms.

## Supporting information

**S1 Appendix.**
(PDF)

## Acknowledgments

We want to thank Andrea Aejmelaeus-Lindström, Anna Blomkvist, and Elmeri Syrjänen for helping with the data collection for the experiments included in this manuscript.

## Author Contributions

**Conceptualization:** Marta Zakrzewska, Marco Tullio Liuzza, Jonas K. Olofsson.

**Formal analysis:** Marta Zakrzewska.

**Funding acquisition:** Marco Tullio Liuzza, Jonas K. Olofsson.

**Methodology:** Marta Zakrzewska.

**Supervision:** Marco Tullio Liuzza, Jonas K. Olofsson.

**Visualization:** Marta Zakrzewska.

**Writing – original draft:** Marta Zakrzewska.

**Writing – review & editing:** Marta Zakrzewska, Marco Tullio Liuzza, Jonas K. Olofsson.

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
