## [Decision Letter · Decision Letter 0]

4 Aug 2022

PONE-D-22-17374Body odor disgust sensitivity (BODS) is related to extreme odor valence perceptionPLOS ONE

Dear Dr. Zakrzewska,

Thank you for submitting your manuscript to PLOS ONE. As you can see, I have sent your manuscript to two independent reviewers who were both positive about it but also raised some important theoretical and methodological issues. Therefore, I invite you to submit a revised version of the manuscript that addresses the points raised during the review process.

We look forward to receiving your revised manuscript.

Kind regards,

Maria Serena Panasiti

Academic Editor

PLOS ONE

Journal Requirements:

2. Please provide additional details regarding ethical approval in the body of your manuscript. In the Methods section, please ensure that you have specified the name of the IRB/ethics committee that approved your study.  

3. Please provide additional details regarding participant consent. In the Methods section, please ensure that you have specified (1) whether consent was informed and (2) what type you obtained (for instance, written or verbal). If your study included minors, state whether you obtained consent from parents or guardians. If the need for consent was waived by the ethics committee, please include this information.

Reviewers' comments:

Reviewer's Responses to Questions

**Comments to the Author**

1. Is the manuscript technically sound, and do the data support the conclusions?

Reviewer #1: Yes

Reviewer #2: Yes

2. Has the statistical analysis been performed appropriately and rigorously? 

Reviewer #1: Yes

Reviewer #2: Yes

3. Have the authors made all data underlying the findings in their manuscript fully available?

Reviewer #1: Yes

Reviewer #2: Yes

4. Is the manuscript presented in an intelligible fashion and written in standard English?

Reviewer #1: Yes

Reviewer #2: Yes

5. Review Comments to the Author

Reviewer #1: The aim of the current study was to investigate how BODS relates to perceptual ratings of pleasant and unpleasant odors. The authors aggregated data from 4 experiments: valence and intensity ratings were collected. Across experiments, they show that individuals with higher BODS levels perceived smells as more highly valenced overall: unpleasant smells were rated as more unpleasant, and pleasant smells were rated as more pleasant. These results suggest that body odor disgust sensitivity is associated with a broader pattern of affect intensity which causes stronger emotional responses to both negative and positive odors.

Overall, the study is interesting and well-written. Introduction is clear and it nicely presents aims and different hypotheses, methods are clear, analyses and results appropriate. I have just some minor comments/suggestions on methods and discussion:

- The aggregated data came from experiments that applied quite different methodology, in particular on odor presentation. This may have affected the related ratings. How the authors dealt with these possible differences? Did the authors perform a comparison of the standardized data between experiments? If there is a difference, could it have affected final results? Please clarify

- Line 157: “Perceptual ratings were done before and after the task.” In this case, which ratings were included in the analyses? Only one data per participant, an average of the repetition or multiple data?

- Please include a table of Means/SD of all key variables (pleasantness intensity ratings of each odor, BODS) per experiment

- In general, the discussion is very long and quite difficult to follow. I suggest the author simplify some parts. For example, the paragraph starting at line 384 is very long and not well connected with the rest of the discussion.

- Line 384 “Our converge with those of a recent study by Tybur and colleagues”: I think this sentence is missing a word

- Line 404 “Our perceptual rating results suggest that people with higher BODS levels are more likely to have a lower criterion i.e., less is needed for them to judge an odor as unpleasant or pleasant, rather than then suggesting that they detect the odor more easily...” This sentence is unclear

Reviewer #2: In the current study, the authors combined previously acquired data including olfactory valence and intensity ratings in addition to Body Odor Disgust Sensitivity (BODS) scores. The main result of the study is a relationship between BODS score and pleasantness scores, independent of intensity. Participants with higher BODS scores report more extreme valence ratings in both negative and a positive direction.

As a general impression the study design is well thought through and the manuscript is very well written.

There are some points which should be addressed before publication.

General comments:

1. The authors describe the negative odors as “pathogen-related cues” and the positive odors as “health-related”. While the pathogen association seems quite strong based on the stimuli chosen, the positive/health related connection is still rather indirect and feels more like speculation; e.g. “lemon” could be associated with a lot of non-health related concepts too. Is there a previous study which validates this connection for the current stimuli?

2. The results reported show relatively small effect sizes; the model for valence ratings including BODS barely passes the 2 SE criterion compared to the Odor model. The interpretation and discussion of the results should reflect the size of the result more. A single sentence at the end of the conclusion does not quite acknowledge this.

a. As an additional note to this. In some following comments I question whether certain additional factors should be considered (e.g. gender interaction and study number). Since the results are so small it becomes imperative to show that these factors do not bias the results.

3. As a general note, the manuscript seems to focus more on model comparison than on the contents of the models. While I personally learned a lot about Bayesian parameter estimation by reviewing this manuscript, I assume that the connection between BODS and valence ratings is the take-home message here. Reporting all model comparisons in a single separate paragraph in the results would allow the authors to stress their findings more in the other sections.

Major comments:

Line 200: Was a potential interaction of gender with the other factors explored? In addition to gender main effects, previous studies have shown that the influence of some factors on olfactory perception depends on the participant’s gender. These potential interactions should be addressed and tested for.

Line 264: Please describe the standardization process and its validity for comparing the different scales used in the studies. Did the data obtained in the different studies present similar distributions?

• On a related note: Was the study in which each data point was obtained tested as a potential confound? Especially the different stimulation methods used in the studies could bias results. Additionally, the different stimulus presentation techniques should be listed as a limitation of the study.

Line 268: The formatting of the results section makes the manuscript difficult to navigate: Both sections 3.1 and 3.2 contain just a single sub-paragraph, questioning the point of using sub-paragraphs. The sub-title for 3.2.1 can simply be removed.

Section 3.3 discusses odor valence ratings, which is the title of section 3.1. The text would be easier to follow if this was moved up and described as section 3.1.2, which would also validate the use of a sub-paragraph for 3.1.1

Tables 2/3/4: Due to their format, these can be difficult to interpret: Rather than looking at a model and checking whether alternatives are better, the reader has to find the best model and verify that the alternatives are worse (WAIC/SE values are shown in the form of increases rather than decreases). To improve readability I suggest highlighting the best fitting model in each of the tables in some way (e.g. bold/italics).

Line 335: Description of table 4 refers to a), b), and c), though this summation isn’t used as such in the table. I suggest simply removing the letters. Arguably, the entire specification could be left out since stating that “BODS internal” refers to the “internal subscale of the BODS” seems a bit unnecessary.

Minor comments:

Line 176: Including a citation in a header feels odd. Adding it to the first mention of the questionnaire in the paragraph improves readability.

Line 397: Random period mid-sentence.

Line 401/402: Phrasing of the parameters is inconsistent. I suggest changing either the first to “how sensitive are they to…” or the second to “what their criterion is....”

Line 416: Since odor detection threshold was not measured, wouldn’t this make the “conclusion” regarding this point a “hypothesis” instead?

Figures: The figure captions don't describe the contents of the image sufficiently. What does "Shaded area and numbers in square" mean?

6. PLOS authors have the option to publish the peer review history of their article (what does this mean?). If published, this will include your full peer review and any attached files.

Reviewer #1: No

Reviewer #2: No

---

## [Author Response · Author response to Decision Letter 0]

26 Sep 2022

We would like to thank the Editor and the Reviewers for their work and insightful comments and suggestions. We did our best to answer each question and incorporate the suggestions into the manuscript. Detailed answers to all comments are uploaded in a separate document.

---

## [Decision Letter · Decision Letter 1]

23 Nov 2022

PONE-D-22-17374R1Body odor disgust sensitivity (BODS) is related to extreme odor valence perceptionPLOS ONE

Dear Dr. Zakrzewska,

Thank you for submitting your manuscript to PLOS ONE.  As you will see, both reviewers are very happy with the changes you have made in your previous submission. However, Reviewer 2 has raised one final concern that needs to be addressed before proceeding with publication. Therefore, I invite you to submit a revised version of the manuscript that addresses this final remark. Please submit your revised manuscript by Jan 07 2023 11:59PM. If you will need more time than this to complete your revisions, please reply to this message or contact the journal office at plosone@plos.org. Please include the following items when submitting your revised manuscript:A rebuttal letter that responds to each point raised by the academic editor and reviewer(s). You should upload this letter as a separate file labeled 'Response to Reviewers'.A marked-up copy of your manuscript that highlights changes made to the original version. You should upload this as a separate file labeled 'Revised Manuscript with Track Changes'.An unmarked version of your revised paper without tracked changes. You should upload this as a separate file labeled 'Manuscript'.If applicable, we recommend that you deposit your laboratory protocols in protocols.io to enhance the reproducibility of your results. Protocols.io assigns your protocol its own identifier (DOI) so that it can be cited independently in the future. For instructions see: https://journals.plos.org/plosone/s/submission-guidelines#loc-laboratory-protocols. Additionally, PLOS ONE offers an option for publishing peer-reviewed Lab Protocol articles, which describe protocols hosted on protocols.io. Read more information on sharing protocols at https://plos.org/protocols?utm_medium=editorial-email&utm_source=authorletters&utm_campaign=protocols.

We look forward to receiving your revised manuscript.

Kind regards,

Maria Serena Panasiti

Academic Editor

PLOS ONE

Journal Requirements:

Reviewers' comments:

Reviewer's Responses to Questions

**Comments to the Author**

1. If the authors have adequately addressed your comments raised in a previous round of review and you feel that this manuscript is now acceptable for publication, you may indicate that here to bypass the “Comments to the Author” section, enter your conflict of interest statement in the “Confidential to Editor” section, and submit your "Accept" recommendation.

Reviewer #1: All comments have been addressed

Reviewer #2: (No Response)

2. Is the manuscript technically sound, and do the data support the conclusions?

Reviewer #1: Yes

Reviewer #2: Yes

3. Has the statistical analysis been performed appropriately and rigorously? 

Reviewer #1: Yes

Reviewer #2: No

4. Have the authors made all data underlying the findings in their manuscript fully available?

Reviewer #1: Yes

Reviewer #2: Yes

5. Is the manuscript presented in an intelligible fashion and written in standard English?

Reviewer #1: Yes

Reviewer #2: Yes

6. Review Comments to the Author

Reviewer #1: I thank the authors for taking the time to revise the manuscript according to previous comments. My concerns have been satisfied. I believe that the MS is now much stronger and worthy of publication in Plos One.

Reviewer #2: The authors made corrections or offered good explanations for most of my comments. However, the (potential) interaction effect of gender feels insufficiently addressed.

While gender affects BODS, I agree that it is unlikely that BODS itself interacts with gender regarding its influence on hedonic ratings. Similarly, I agree that for odor intensity an interaction is not as relevant either.

However, pleasantness ratings are obtained regarding odorants which are specifically categorized as pleasant or unpleasant. This is problematic since, by definition, the pleasantness ratings are drawn to opposite ends of the rating spectrum: A correctly labeled positive odor will naturally be rated as pleasant and a correctly labeled negative odor will be rated unpleasant. Due to this innate property of hedonic ratings interactions are more logical than main effects when using categorized stimuli.

Reading the paragraph starting at line 304 (or a similar one at 332) shows how odd it is to look at just the main effect. The authors specify that individuals with higher BODS rate unpleasant smells less pleasant and pleasant smells more pleasant. Then the next sentence specifies that women rate “the smells” as more pleasant. Surely, most readers will instantly question whether this refers to the pleasant or the unpleasant smells.

Even if this interaction was not part of the initial hypothesis, verifying whether it affects the optimal model is important.

Minor comment: The terms "odor" and "smell" are both used. I can't tell if there is a specific distinction made when one or the other is used or whether it is just to avoid repetition. At times it felt a bit odd to read.

7. PLOS authors have the option to publish the peer review history of their article (what does this mean?). If published, this will include your full peer review and any attached files.

Reviewer #1: No

Reviewer #2: No

---

## [Author Response · Author response to Decision Letter 1]

26 Jan 2023

Dear Editor,

Again, we would like to thank the Editor and the Reviewers for their work and insightful comments and suggestions. Below we provide a summary of changes in this new version of our manuscript.

Best regards,

Authors

Response to Reviewers’ comments:

1. We have addressed the final concern raised by Reviewer #2. 

We realized that the Reviewer has a good point it highlighting that ‘verifying whether [the interaction] affects the optimal model is important’. We performed additional analyses in which we investigated potential gender differences in the effects of BODS on valence ratings (also looking separately at pleasant and unpleasant odors), as well as any potential interactions between gender, BODS and odor category. Similarly, we looked into gender differences in intensity ratings. These exploratory analyses and their results are brought up in the main text and described in detail in Supplementary materials. We hope that our analysis help clarify the potential differences between men and women.

Relevant text in the main manuscript: 

- Methods section: 2.5.3 Exploratory analysis of gender differences (p. 8 lines 288 – 291)

- Results: “Exploratory analysis suggests that this gender difference might apply mostly to pleasant odors (see Supplementary materials, 2.1. Gender differences, body odor disgust sensitivity and valence ratings)”(p. 10, lines 358 – 360)

- Results: “The exploratory analysis did not provide any more insight into gender differences (see Supplementary materials, 2.2. Gender differences in intensity ratings)” p.11 lines 415 – 417)

2. We did minor updates to the text itself in order to correct previously unseen linguistic mistakes, and to improve the reading experience.

3. We realized we made a mistake when describing gender differences in intensity ratings in the previous submission, saying that it was women who rated smell as somewhat less intense, while in fact, it was men who rates smells as less intense than women. This effect is correctly reported in the current submission. (p. 11, line 416)

---

## [Decision Letter · Decision Letter 2]

30 Mar 2023

Body odor disgust sensitivity (BODS) is related to extreme odor valence perception

PONE-D-22-17374R2

Dear Dr. Zakrzewska,

We’re pleased to inform you that your manuscript has been judged scientifically suitable for publication and will be formally accepted for publication once it meets all outstanding technical requirements.

Kind regards,

Maria Serena Panasiti

Academic Editor

PLOS ONE

Reviewers' comments:

Reviewer's Responses to Questions

**Comments to the Author**

1. If the authors have adequately addressed your comments raised in a previous round of review and you feel that this manuscript is now acceptable for publication, you may indicate that here to bypass the “Comments to the Author” section, enter your conflict of interest statement in the “Confidential to Editor” section, and submit your "Accept" recommendation.

Reviewer #1: All comments have been addressed

Reviewer #2: All comments have been addressed

2. Is the manuscript technically sound, and do the data support the conclusions?

Reviewer #1: Yes

Reviewer #2: Yes

3. Has the statistical analysis been performed appropriately and rigorously? 

Reviewer #1: Yes

Reviewer #2: Yes

4. Have the authors made all data underlying the findings in their manuscript fully available?

Reviewer #1: Yes

Reviewer #2: Yes

5. Is the manuscript presented in an intelligible fashion and written in standard English?

Reviewer #1: Yes

Reviewer #2: Yes

6. Review Comments to the Author

Reviewer #1: I thank the authors for taking the time to revise the manuscript according to previous comments. My concerns have been satisfied. I believe that the MS is now much stronger and worthy of publication in PLOS ONE.

Reviewer #2: (No Response)

7. PLOS authors have the option to publish the peer review history of their article (what does this mean?). If published, this will include your full peer review and any attached files.

Reviewer #1: No

Reviewer #2: No

---

## [Editor Report · Acceptance letter]

12 Apr 2023

PONE-D-22-17374R2 

Body odor disgust sensitivity (BODS) is related to extreme odor valence perception 

Dear Dr. Zakrzewska:

I'm pleased to inform you that your manuscript has been deemed suitable for publication in PLOS ONE. Congratulations! Your manuscript is now with our production department. 

Kind regards, 

on behalf of

Dr. Maria Serena Panasiti 

Academic Editor

PLOS ONE